# Baicalin Represses Type Three Secretion System of *Pseudomonas aeruginosa* through PQS System

**DOI:** 10.3390/molecules26061497

**Published:** 2021-03-10

**Authors:** Pansong Zhang, Qiao Guo, Zhihua Wei, Qin Yang, Zisheng Guo, Lixin Shen, Kangmin Duan, Lin Chen

**Affiliations:** 1Key Laboratory of Resource Biology and Biotechnology in Western China, Ministry of Education, Northwest University, Xi’an 710069, China; zps102065106@stumail.nwu.edu.cn (P.Z.); 201820891@stumail.nwu.edu.cn (Z.W.); 201931986@stumail.nwu.edu.cn (Q.Y.); zishengguo@nwu.edu.cn (Z.G.); shenlx@nwu.edu.cn (L.S.); 2College of Natural Resources and Environment, Northwest A&F University, Yangling 712100, China; shuiyiwei83@163.com; 3Department of Oral Biology & Medical Microbiology & Infectious Diseases, Rady Faculty of Health Sciences, University of Manitoba, 780 Bannatyne Ave., Winnipeg, MB R3E 0W2, Canada

**Keywords:** *Pseudomonas aeruginosa*, virulence factors, baicalin, Type III secretion system, PQS inhibition, mouse lung infection model

## Abstract

Therapeutics that target the virulence of pathogens rather than their viability offer a promising alternative for treating infectious diseases and circumventing antibiotic resistance. In this study, we searched for anti-virulence compounds against *Pseudomonas aeruginosa* from Chinese herbs and investigated baicalin from *Scutellariae radix* as such an active anti-virulence compound. The effect of baicalin on a range of important virulence factors in *P. aeruginosa* was assessed using *luxCDABE*-based reporters and by phenotypical assays. The molecular mechanism of the virulence inhibition by baicalin was investigated using genetic approaches. The impact of baicalin on *P. aeruginosa* pathogenicity was evaluated by both in vitro assays and in vivo animal models. The results show that baicalin diminished a plenty of important virulence factors in *P. aeruginosa*, including the Type III secretion system (T3SS). Baicalin treatment reduced the cellular toxicity of *P. aeruginosa* on the mammalian cells and attenuated in vivo pathogenicity in a *Drosophila melanogaster* infection model. In a rat pulmonary infection model, baicalin significantly reduced the severity of lung pathology and accelerated lung bacterial clearance. The PqsR of the *Pseudomonas* quinolone signal (PQS) system was found to be required for baicalin’s impact on T3SS. These findings indicate that baicalin is a promising therapeutic candidate for treating *P. aeruginosa* infections.

## 1. Introduction

Rather than targeting viability, an alternative approach focusing on the functions that are essential for pathogenicity offers a new way for therapeutic development to control infectious diseases. Anti-virulence therapeutics potentially have additional advantages over traditional viability-targeted antibiotics, which include reduced risk of antibiotic resistance in pathogens due to the lack of or decreased selective pressure from these agents and the preservation of the host endogenous microbiome.

*P. aeruginosa* is frequently found and can cause various infections in humans including bloodstream infection, pneumonia and urinary tract infection, and infection in burn patients. Pulmonary infection by *P. aeruginosa* in cystic fibrosis patients is attributed to the major cause of pulmonary failure and mortality in these patients. With decades of conventional antibiotic usage, increasing occurrence of multi-drug resistant *P. aeruginosa* has led to enormous difficulties in clinical treatment. The successful infection of *P. aeruginosa* in various hosts is considered due to the plenty of virulence factors such as Type III secretion systems (T3SS), motility, proteases and exopolysaccharides. T3SS plays an essential role in acute infection which injects toxic effector proteins into eukaryotic host cells [1,2,3]. T3SS has also become a promising target of anti-virulence therapy. PcrV is an important translocation protein of the T3SS in *P. aeruginosa* [4]. In a mouse lung infection model with PcrV protein immunization, protection against lethal lung infection, lung injury, and cellular toxicity appeared to be mediated by PcrV antibodies [5,6]. Vaccination against PcrV is being developed and has good efficacy in treating acute infection caused by *P. aeruginosa*.

In *P. aeruginosa*, overwhelming evidence indicates that numerous virulence factors are controlled transcriptionally by global regulatory quorum-sensing (QS) systems [7]. There are three density-dependent cell-to-cell communication circuits in *P. aeruginosa*: Two of them are based on the synthesis and detection of acyl homoserine lactone (AHL) termed *las* and *rhl* systems, and the third is the 2-alkyl-4(1H)-quinolone (AHQ) signal-based system [8,9]. The major AHQ signals include 2-heptyl-3-hydroxy-4-quinolone, known as the *Pseudomonas* quinolone signal (PQS), and the immediate precursor of PQS, 2-heptyl-4-quinolone (HHQ) [10,11,12]. The expression of the PQS operons (including *pqsABCDE* and *phnAB* genes) and the HHQ production are positively regulated by the LysR type transcriptional regulator PqsR. The flavin-dependent monooxygenase encoded by the *pqsH* gene converts HHQ to PQS. In addition, global regulators such as QS systems and GacS/GacA have been proved to play a dominant role in modulating the expression of T3SS [13,14,15]. These virulence controlling systems present an ideal anti-virulence target in *P. aeruginosa.*

Chinese herbs have been used for over a thousand years and the usage of herb extracts in treating infectious diseases is well documented. Many compounds from herbs have been identified as effective in the treatment of different human diseases [16,17,18]. Hence, they may represent a potential resource for anti-virulence compounds. *Scutellariae radix* (*Scutellaria baicalensis* Georgi), belonging to the class of herbs with functions of Qing Re Jie Du (i.e., treating symptoms resembling infections), has been wildly used for treatment of human diseases such as respiratory tract infection, tracheitis, abscess and sores in China. Previous studies have focused on the antibacterial ability of *Scutellariae radix* and the identification of its constituents. Baicalin, as one of the major bioactive flavone constituents in *Scutellariae radix*, has been reported to exhibit a range of pharmacologic actions such as antioxidant activity and antitumor effects [19]. Seemingly contradicting results have been reported regarding whether baicalin has antibacterial activity against pathogens such as *P. aeruginosa*, probably due to the fact that baicalin shows antibacterial activity only at high concentrations (i.e., >1 mg/mL) [20,21]. Recent studies indicate that baicalin is an AHL-based QS inhibitor against *P. aeruginosa* and *Burkholderia cepacia* complex [22] and it reduces *P. aeruginosa* biofilm formation and *las* and *rhl* systems controlled virulence factors including elastase, LasA protease, pyocyanin and rhamnolipids [23]. While these studies show that baicalin is an inhibitor of the AHL-mediated QS in *P. aeruginosa*, its effect on other important virulence factors such as T3SS and the PQS-mediated QS system remained unclear. In addition, the therapeutic potential of baicalin for treating pulmonary infections has yet to be tested, despite the fact that *P. aeruginosa* causes pneumonia and its persistent lung infections causes chronic inflammation and destruction of lung tissue, eventually leading to death in the majority of patients with cystic fibrosis [24,25,26]. *P. aeruginosa* infection has also been identified in the lungs of COVID-19 patients [27].

In this study, we investigated the effect of baicalin, an active compound in *Scutellariae radix*, on numerous virulence factors in *P. aeruginosa* and analyzed its anti-pathogenic impact using cell cultures and two animal models including a rat pulmonary chronic infection model. We report here the results showing baicalin treatment reduced the cellular toxicity of *P. aeruginosa* on the mammalian cells and attenuated in vivo pathogenicity in both fruit fly infection model and a rat pulmonary infection model. Both reduced inflammatory response and enhanced bacterial clearance were observed in the host treated with baicalin. With a focus on the particularly important virulence factor in *P. aeruginosa*, T3SS, we examined the impact of baicalin on T3SS and revealed the underlying molecular mechanism. We report that the inhibition activity on the T3SS by baicalin was via PqsR of the *Pseudomonas* Quinolone Signal (PQS) system. These findings indicate that baicalin not only inhibits AHL-mediated QS but also PQS system and T3SS, supporting that baicalin represents a promising antipathogenic therapeutic candidate for treatment of *P. aeruginosa* infection.

## 2. Results

### 2.1. The Effect of Crude Extracts on Virulence Factors in P. aeruginosa PAO1

We initially screened for virulence inhibitors from herbal medicines which are known as the functions of Qing Re Jie Du (i.e., treating symptoms resembling infections). The expression of virulence related genes was monitored using expression monitoring assay as described in materials and methods [28] and the effect on growth was reflected by clear halo in the double agar diffuse assay. The ethanolic extract of *Scutellariae radix* could inhibit the expression of virulence factors including *lasI*, *lasR*, *rhlI*, *rhlR*, *fliC*, *aprA*, *xcpR*, *migA*, *exoT*, *exsD*, *rnr*, *exoS*, *phzA2*, *oprH* and *pilG*, whereas induced the expression of *phzA1* (data not shown). The inhibition of the ethanolic extract of *Scutellariae radix* on the expression *lasR*, *rhlI*, *lasR* and *exoS* was presented in Figure 1A. Similarly, the aqueous extract of *Scutellariae radix* inhibited the expression of *lasI*, *lasR*, *rhlI*, *rhlR*, *fliC*, *aprA*, *xcpR*, *migA*, *exoS*, *exoT*, *exsD*, *rnr*, *oprH* and *pilG*, while there was no observable inhibition of the expression of *phzA1* and *phzA2*. The serial dilutions aqueous and ethanolic extract of *Scutellariae radix* had no effect on the growth of the *P. aeruginosa*, since there were no zones of inhibition appeared.

### 2.2. The Extract of Scutellariae Radix Reduces Swarming Motility, but Has No Effect on Swimming and Twitching Motilities

It has been demonstrated that QS system, surface wetting agent rhamnolipid, type IV pili and the flagellum all contribute to swarming motility [29,30]. Since the aqueous and ethanolic extracts of *Scutellariae radix* inhibited the expression of quorum sensing related genes, rhamnolipid synthetic gene and flagellar biosynthesis gene, then we determined the effect of crude extracts on swarming motility respectively. As shown in Figure 1B, the ethanolic extract of *Scutellariae radix* could apparently attenuate the swarming motility of *P. aeruginosa* PAO1. However, the aqueous extract exhibited minor repression on the swarming motility of PAO1. Swimming and twitching are two other types of motilities of *P. aeruginosa*. The effect of *Scutellariae radix* extract on those motilities was also determined. Both the ethanolic extract and aqueous extract of *Scutellariae radix* had no influence on twitching and swimming motilities of *P. aeruginosa* PAO1 (data not shown). 

### 2.3. Baicalin Inhibits the Expression of Virulence Factors in P. aeruginosa 

The regulation of *Scutellariae radix* on the virulence related genes expression suggested that there may be one or some active anti-virulence compound(s) existed in the crude extract of *Scutellariae radix*. Baicalin (Figure 2A) is one of the major constituents in *Scutellariae radix* and has been used to treat respiratory tract infection, tracheitis, carbuncle, abscess and sores. The antibacterial property of baicalin is weak, however, since the MIC range of baicalin against clinically common pathogenic bacteria *P. aeruginosa* is 0.4–12.5 mg/mL and the MBC range is 0.8–25 mg/mL [31]. We speculated that baicalin might target other aspect of pathogenic bacteria such as virulence factor rather than focusing on its viability to treat infectious diseases. To test such a possibility, we examined the effect of baicalin on the expression of virulence related genes using *luxCDABE*-based reporter fusion system. The effect of baicalin on the bacterial growth was monitored and the results demonstrated that baicalin had no influence on the growth of *P. aeruginosa* PAO1 even at the concentration of 1 mg/mL. However, baicalin significantly repressed the expression of a plenty of important virulence factors in *P. aeruginosa* such as T3SS genes *exoS*, *exoY*, *exoT*, *exsC* and *exsD*; quorum sensing genes *lasI*, *lasR*, *rhlI*, *rhlR*, *pqsA* and *pqsR;* motility related genes *fliC* and *pilG*; globe regulators *rsmA*, *rpoS*, *vfr* and *gacA;* and *lasB*, *rhlA* (Appendix A). Interestingly and unexpectedly, the expression of phenazine synthesis genes *phzA1* and *phzA2* were induced by baicalin. The inhibition of *exoS*, *exoT* and *exsD* expression by baicalin is presented in Figure 2B. 

The zinc metalloproteases elastase, encoded by *lasB* gene, can cause host tissue degradation and contribute to the pathogenesis of *P. aeruginosa* [32]. Rhamnolipid is expressed by *rhlA* gene and plays important roles in motility and the maintenance of biofilm architecture. The result obtained above showed that baicalin inhibited the expression of *lasB* and *rhlA*. Thus, we examined the effect of baicalin on *P. aeruginosa* motilities and the production of elastase and rhamnolipid. As shown in Figure 2C, swarming motility was impaired in the presence of baicalin (250 μg/mL) compared with the control. Similar to the crude extract of *Scutellariae radix*, baicalin had no impact on the swimming and twitching motilities. In agreement with the decreased gene expression, baicalin significantly inhibited the production of elastase at the concentration of 250 μg/mL (Figure 2D,E).

### 2.4. Reduction of the Cellular Toxicity of PAO1 on the Mammalian Cell by Baicalin

T3SS is one of central virulence in *P. aeruginosa*, enabling the bacteria to penetrate eukaryotic cells and cause the cellular toxicity of PAO1 on the mammalian cell [1,2,3]. Since baicalin inhibited the expression of *P. aeruginosa* T3SS genes, including secreted effectors encoding genes *exoS*, *exoT*, and *exoY*, we tested whether T3SS repression by baicalin has a biological effect in *P. aeruginosa* cytotoxicity. The LDH release assay was performed to determine the cytotoxicity of baicalin-treated bacteria on the murine mammary carcinoma cells EMT6. As presented in Figure 3, in the presence of baicalin, the LDH release of EMT6 cells infected by *P. aeruginosa* PAO1 was much lower than cells without baicalin, demonstrating that baicalin could reduce the cytotoxicity of *P. aeruginosa*.

### 2.5. Baicalin Attenuates the Pathogenicity of PAO1 in the Fruit Fly Model

*Drosophila melanogaster* has been used as a model to analyze *P. aeruginosa* virulence and interactions between this bacterium and innate host defense. The fly infection model was adopted to test the effect of baicalin on *P. aeruginosa* PAO1 pathogenicity. Flies were fed with PAO1 on solidified 5% sucrose agar containing 250 μg/mL baicalin or DMSO as control, and fly survival was monitored daily. The data showed a significant increasing in fly survival (*p* = 0.00014 < 0.001) when treated with baicalin compared with that in the control group (Figure 4), clearly indicating that baicalin could attenuate the pathogenicity of *P. aeruginosa* in the fly infection model.

### 2.6. Baicalin Attenuates the Pathogenicity of P. aeruginosa PAO1 and Host Inflammatory Response in the Rat Pulmonary Infection Model

Considering the reduction of cytotoxicity on mammalian cell and the attenuation of pathogenicity in fly infection model by baicalin, we further determined the effect of baicalin on *P. aeruginosa* pathogenicity in a pulmonary infection model. Forty-five rats were inoculated with alginate beads of *P. aeruginosa* PAO1 and fifteen rats were inoculated with alginate beads of saline. On the second day after challenge, four rats infected by *P. aeruginosa* died and thirty-nine of forty-one rats remained were randomized into three groups. Each group contained 13 rats, and they were Baicalin-treated group, Cefepime-treated group and (Untreated) Model group respectively. All rats inoculated with alginate beads of saline were normal in the blank control group. Rats in each group received drugs or placebo once a day from the second day after challenge.

During the infection stage, there were five rats: three in Model group, one in the Baicalin-treated group and one in the Cefepime-treated group died. Animals in all groups were sacrificed on day 4 post infection. As shown in the Figure 5 and Table 1, macroscopic pathology of lungs in the Baicalin-treated group as well as the Cefepime-treated group were significantly milder than the Model group respectively (*p* = 0.02 and *p* = 0.01). The lungs of eight rats in Baicalin-treated group and nine in the Cefepime-treated group had returned to normal macroscopically. However, all of lungs in Model group exhibited different degrees of pathologic change. One rats with score of 3, and two rats with score of 2 were found in Baicalin-treated group, while there were three rats with score of 2 in Cefepime-treated group. The adherences, hemorrhages, abscesses (>1 by 2mm), and atelectasis (>3 mm) with score of 4 were only found in rats of the Model group (Figure 5A,B).

Lung sections were examined by H&E staining to assess tissue damage resulting from PAO1 infections. As presented in the Figure 6, lung tissues in Baicalin-treated group as well as Cefepime-treated group showed less inflammatory cell infiltration and milder lung haemorrhage compared with the Model group. In addition, although rats in control group were not infected by bacterium, the lungs still showed light haemorrhage due to the inoculation of saline in alginate beads.

All animals were anaesthetized and blood was drawn from abdominal aorta. Concentrations of TNF-α were determined by enzyme-linked immunosorbent assay (ELISA) using kit. The result showed that the TNF-α levels in rats of Baicalin-treated group as well as Cefepime-treated group obviously lower than the Model group respectively (*p* = 0.027, 0.009 < 0.05). Despite that a significant reduction of TNF-α level in rats of control group compared with the Model group, we observed that rats in Cefepime-treated group also had much lower TNF-α level than the control group (*p* = 0.041 < 0.05) (Figure 7).

In concordance with the results of the macroscopic lung pathology study, lower value of the lung bacterial colony forming units (CFU) was also observed in the rats treated intravenously with baicalin (Table 2). Likewise, the lung bacterial CFU in the Cefepime-treated group was significantly lower than that in Model group. 8 of 12 rats in the Baicalin-treated group had cleared pathogenic bacteria PAO1. Bacterial count of lung in one rat was fewer than 10 CFU/mL and other two rats were found to be infected with 10^2^ and 10^3^ CFU/mL respectively. In the Model group, bacterial counts of four rats were found to be higher than 10^4^ CFU/mL, two rats were infected with higher than 10^5^ CFU/mL and the rest four rats were infected with higher than 10^2^ CFU/mL. In the Cefepime-treated group, 9 of 12 rats had cleared the pathogenic bacteria PAO1 and other three rats were found to be infected with less than 10^2^ CFU/mL. The incidence of rats with less than 10^2^ CFU/mL in Baicalin-treated group or Cefepime-treated group was significantly lower than that in the Model group. In conclusion, all these observations demonstrated that baicalin could assist bacterial clearance by the host, and reduce the lung pathology in the pulmonary infection model, further confirming the attenuation of *P. aeruginosa* pathogenicity by baiclian.

### 2.7. Baicalin Exerted Its Impact on T3SS via PqsR of the PQS System 

In view of the above results that baicalin influenced a range of virulence factors, especially T3SS, and attenuated the in vivo and in vitro pathogenicity of *P. aeruginosa* PAO1 in different animal models, we attempted to investigate the mechanism of action of baicalin. Since T3SS is one of the major virulence factors in *P. aeruginosa,* we explored the mechanism that baicalin repressed T3SS using *exoS* as a reporter. Several global regulators have been documented to modulate the expression of T3SS. We examined the effect of baicalin on *exoS* in T3SS regulator mutants, including *gacA*, *rhlI*, *pqsR*, and *pqsH* mutants. The results indicate that the effect of baicalin on the *exoS* expression in both *gacA* and *rhlI* mutant remained same as in the wild type (data not shown), suggesting neither the *gac/rsm* pathway nor the *rhl* QS system was the pathway through which baicalin inhibited T3SS.

Our previous study showed that the T3SS is negatively regulated by the PQS system in *P. aeruginosa* [33]. To investigate whether the effect of baicalin on the virulence factors was depended on the PQS system, we tested the effect of baicalin on *exoS* in *pqsR* and *pqsH* mutants. The results obtained show that the inhibition of *exoS* by baicalin disappeared in the *pqsR* mutant (Figure 8). In agreement with previous observation, the *exoS* expression was higher in both mutants and the addition of PQS in *pqsH* mutant restored the effect of baicalin on *exoS* but not in the *pqsR* mutant (Figure 8). The results indicate that the inhibition of T3SS by baicalin required both PQS and PqsR, i.e., the PQS system. 

## 3. Discussion

An emerging approach for antimicrobial drug development is to target functions essential for infection, instead of targeting viability. Such an approach has several potential advantages including the preservation of the host endogenous microbiome and exerting less selective pressure on the pathogens [34]. *P. aeruginosa* is an opportunistic pathogen associated with seriously acute and chronic infections in humans. Extensive studies have focused on screening for the compounds inhibiting virulence factors in *P. aeruginosa*, including inhibitors targeting T3SS [35,36,37] and QS system [38,39,40,41], which play a crucial role in *P. aeruginosa* pathogenicity. 

Our screen of Chinese herbs that are known for the functions of Qing Re Jie Du (i.e., treating symptoms resembling infections) for virulence inhibitors revealed that *Scutellariae radix* extract inhibited *P. aeruginosa* virulence factors, but had no influence on cell viability. In this study, we show that baicalin, one of major flavonoids isolated from *Scutellariae radix* can significantly reduce the virulence factors such as the production of rhamnolipids, elastase, swarming motility, and interestingly T3SS. Our data suggest that the repression of T3SS was via the PQS system. This is an addition to the reported effect of baicalin on AHL-mediated QS systems and their controlled virulence factors. 

*P. aeruginosa* T3SS gives rise to death to many mammalian cell-types by directly injecting cytotoxins into the targeted cells. T3SS caused killing is believed to play a pivotal role in the pathogenesis of *P. aeruginosa* [1,42]. Thus, T3SS has been proposed to be an attractive target for curing *P. aeruginosa* infections and many researches have focused on the identification of the T3SS inhibitors from nature source. It is also reported that vaccination against PcrV which is an important translocation protein in T3SS system enables the survival of challenged mice and reduces lung inflammation and injury [5,6]. Molecules inhibiting the activity of T3SS effector or the transcriptional and secretion process have been proved to be effective in animal infection model [35,36,37,43]. Apparently, T3SS is a promising target for treatment of *P. aeruginosa* infectious diseases. The investigation that baicalin can repress T3SS genes such as *exoS*, *exoT*, *exoY*, *exsD* and *exsC*, suggesting that baicalin is a promising drug candidate that targets T3SS in *P. aeruginosa*. 

In agreement with the important role of T3SS in pathogenicity, baicalin reduced the cellular toxicity of PAO1 on the mammalian cell. It significantly attenuated the pathogenicity of PAO1 in fruit fly infection model. The result using a pulmonary infection model demonstrated that baicalin could decrease the inflammatory response, reduced lung damage, and enhance bacterial clearance in the lungs. The therapeutic effect of baicalin in these animal models is likely initiated from its inhibition of both T3SS and the AHL-based QS systems. It is interesting to note although baicalin showed no inhibitory effect on *P. aeruginosa* growth in vitro, the bacterial loads in the experimental animal significantly decreased in the baicalin treated group compared to those in the untreated model group. The decreased virulence factor production and reduced pathogenicity in general probably enabled the host to render better immune responses which clear the bacterium from the tissue. These findings support that baicalin is a promising antipathogenic drug candidate for treating *P. aeruginosa* pulmonary infections. 

Further investigation on the underlying mechanism demonstrated that the inhibition activity on the T3SS by baicalin was dependent on the PQS system in *P. aeruginosa*. Our results indicate that both PQS and its cognate receptor, the transcriptional regulator PqsR, were essential for the inhibitory activity of baicalin on T3SS, clearly indicating that baicalin exerted its repression on T3SS through the PQS system. We previously observed that T3SS is negatively regulated by PQS systems. It is possible baicalin simply enhanced the inhibitory activity of PQS system on T3SS. However, our results showed that baicalin also inhibited the expression of genes of the PQS system, consistent with previous report [23]. We speculate that the repression of the PQS system by baicalin was probably overtaken by its effect on T3SS. The level of PQS gene in the presence of baicalin remained substantial, which was probably sufficient to enable baicalin to exert an inhibitory effect on T3SS. The investigation of the effect of baicalin on the PQS-PqsR interaction may generate further insight on the role of baicalin and will be pursued in the future. 

## 4. Materials and Methods

### 4.1. Bacterial Strains and Culture Conditions

The strains and plasmids are listed in Table 3. All strains were grown at 37 °C on LB (Lubria-Bertani) agar plates or LB liquid medium with shaking at 200 rpm. The concentration of antibiotics was used as follow: Kanamycin (Kn) was used at 50 μg/mL in *Escherichia coli*, and Tetracycline (Tc) and Trimethoprim (Tmp) were used at 300 μg/mL in *P. aeruginosa*. Baicalin (DMSO solution): Baicalin (98% HPLC, XiaoCao) was dissolved in DMSO (Sigma-Aldrich Corporation, Product Number D 2650, Saint Louis, MO, USA), filter-sterilized using 0.22 μm (pore size) Iwaki filter disks and stored at 4 °C for several weeks. Baicalin (PBS solution): Baicalin (0.005 g) was dissolved in 1 mL PBS (pH = 7.0) and sterilized by filtration (0.22 μm) before use. This was prepared just before experiments.

### 4.2. Plant Extraction

The air-dried *Scutellariae radix* were boiled for 2 h with 70% ethanol firstly and then distilled water (*Scutellariae radix* weight to solvent weight 1:5–1:10) respectively. The crude ethanolic extract and aqueous extract were filtered using filter paper. The extracts were concentrated at 40 °C using Buchi vacuum evaporator (Switzerland) and then the lyophilized powder of extracts was conserved in −20 °C. Extracts were dissolved into corresponding solvent (distilled water or 50% methanol) to obtain desired solutions for testing and all solutions were sterilized by Syringe-driven Filters (0.22 μm pore size).

### 4.3. Expression Monitoring Assay

The promoter-reporter fusions of virulence-related genes were constructed as described previously [47]. Virulence genes expression was measured using the reporter vector which contains the promoterless *luxCDABE* operon downstream of the gene promoter [51]. Double agar diffusion assay was adopted to measure the anti-virulence activity of the samples. Bacterial growth inhibition would result in a clear halo around the disc. The reporter strains (as listed in Table 3) were incubated in LB medium at 37 °C overnight with orbital shaking at 200 rpm. The upper layer of LB medium (0.7% agar) was cooled to 40 °C and mixed with 100 μL of the overnight cultures which was adjusted to 0.1 (OD_600_), and subsequently spread on the lower layer of LB medium (1% agar) prepared previously. The sterile paper discs with six millimeters diameter were prepared and were impregnated with the 10 μL serial dilutions of extracts. Ethanol extracts of different plants were dissolved in 50% methanol. Discs loaded with sterile water or 50% methanol alone were used as control. The plates were incubated at 37 °C overnight and the effect of herb extracts on the gene expression was monitored by imaging in a LAS-3000 imaging system (Fuji Corp, Tokoyo, Japan.).

Expression monitoring in soft agar medium was measured as follow. Overnight cultures of the reporter strains were adjusted to an OD_600_ of 0.2. The upper layer of LB medium (0.7% agar), cooled to 40°C, was mixed with 100 μL of the adjusted overnight cultures supplemented with 0.5 mg/mL baicalin or 100 μL of the diluted overnight cultures alone (as control), and then the mixture was spread on the lower layer of LB medium (1% agar) prepared previously. The plates were incubated overnight at 37 °C and imaged by LAS-3000 imaging system (Fuji Corp, Tokoyo, Japan).

An integration plasmid CTX6.1 [44] originating from plasmid mini-CTX-*lux* was used to construct chromosomal fusion reporters as described previously [29].

### 4.4. Motility Assays

The motility assay was performed as described previously with minor modifications [52]. Swimming or swarming motility plates were prepared as described in reference [52]. Briefly, two microliters of cultures were carefully dropped on the swimming or swarming plates supplemented with compounds or an equal volume of solvent as control. The plates were cultured at proper conditions and the results were taken by LAS-3000 imaging system. In the twitching motility assays, freshly LB agar plates were used. Single colony was picked using a sterile toothpick and then stabbed through the agar layer to the bottom of the Petri dish. Photographs were taken with the LAS-3000 imaging system (Fuji Corp, Tokoyo, Japan) after incubation at 37 ℃ for 48 h.

### 4.5. Elastase Assay

LasB protease (elastase) was measured by the elastin Congo red assay with slightly modifications [53]. Briefly, several PAO1 colonies were inoculated into LB medium in the presence of 0.5 mg/mL baicalin or an equal volume of DMSO (as a control) to an OD_600_ of 2.0. An additional control is LB medium with 0.5 mg/mL baicalin which intended to eliminate error caused by the color of baicalin. Cells were collected by centrifugation and the supernatants were sterilized by syringe driven filters (0.22 μm pore size). The samples of 1 mL of supernatant were added to the tubes containing 5 mg elastin Congo red (Sigma, Saint Louis, MO, USA) and 2 mL of 0.1 M phosphate buffer (pH 7.0), and incubated at 37 °C with shaking (200 rpm) for 3 h. After insoluble was removed by centrifugation, the absorbance at an optical density of 495 nm was measured with a spectrophotometer zeroed on sample incubated with medium with 0.5 mg/mL baicalin. The experiments were repeated at least three times. 

### 4.6. Rhamnolipid Assay

Rhamnolipid was measured as described previously [45]. Briefly, cells were grown in Peptone Tryptone Soya Broth. When cultures were reached at mid-exponential-phase, cells were washed by centrifugation and resuspended in modified GS medium (medium group C) [54] which containing baicalin (0.5 mg/mL) or an equal volume of DMSO at an OD_660_ of 0.2. An additional control is modified GS medium with 0.5 mg/mL baicalin which intended to eliminate error caused by the color of baicalin. After incubating at 37 °C for a total of 80 h with shaking (200 rpm), the cultures were centrifuged at 16,000× *g* for 5 min. The supernatants were sterilized using syringe-driven filters (0.22 mm pore-size) and extracted twice using two volumes of diethyl ether. The pooled ether extracts were evaporated to dryness after extracting once with 20 mM HCl. The content of rhamnolipid in each sample was measured by comparing with rhamnose standards via duplicate orcinol assays [55].

### 4.7. Cytotoxicity Measurement

Cytotoxicity of *P. aeruginosa* on eukaryotic cells was measured by monitoring the release of the cytoplasmic enzyme lactate dehydrogenase (LDH). Murine mammary carcinoma cells (EMT6) were routinely grown in RPM1 1640 medium supplemented with 10% fetal bovine serum (FBS) and 2 mM glutamine. Prior to infection, confluent EMT6 cells were washed and resuspended at a concentration of 2 × 10^4^ cells/mL and added onto 96-well plate. PAO1 overnight cultures were sub-cultured into fresh LB, and grown to mid-log phase. EMT6 cells were infected with mid-log-phase *P. aeruginosa* PAO1 at an initial multiplicity of infection (MOI) of 100. After 4 h of infection, the extent of LDH release was assayed with the CytoTox 96 Kit in accordance with the manufacturer’s instructions (Promega, Madison, WI, USA). 

### 4.8. Fly Infections Model

The fly feeding assay was performed as previously described [56]. PAO1 overnight cultures were diluted to an OD_600_ of 0.2 using LB medium. The cell pellet from 1.5 mL of overnight culture was collected by centrifugation and resuspended in 100 μL 5% (*w*/*v*) sucrose. The resuspended cells were spotted onto a sterile filter paper that was placed on the surface of 1.5 mL of solidified 5% sucrose agar containing 250 μg/mL baicalin or an equal volume of DMSO as control in the wells of a 24-well plate. Male Canton S flies (3–5 days old) were starved for 3 h and 11–15 flies were added to each well of the 24-well plate. Carbon dioxide was used for anesthetizing flies throughout the whole process. The plates were placed at 26 °C in a humidity control environment. The number of live flies was documented and counted at 24 h intervals. A minimum of 60 flies were used for each infection group. The data is a representative of at least three independent experiments.

### 4.9. The Rat Pulmonary Infection Model 

The effect of baicalin on the *P. aeruginosa* pathogenicity was assayed in a rat pulmonary infection model by a revised method [57]. We extended the infection stage to test the effect of baicalin in treatment of pulmonary infection. Bacterial was inoculated in alginate beads which cannot be cleared easily by the host. 

Experimental animals used were female, 5-week-old Sprague-Dawley rats (provided by Medical Experimental Center of Xi’an Jiaotong University) with a body weight around 150 g. The rats were housed under specific-pathogen-free conditions. All animal experiments were designed and operated in accordance with the Institutional Animal Care and Use Committee of Xi’an Jiaotong University.

Immobilization of *P. aeruginosa* in seaweed alginate beads was done as described with minor modification [58]. Briefly, one bacterial colony was inoculated into 100 mL LB medium and cultured at 37 °C for 20 h with shaking (200 rpm). Centrifuge the culture at room temperature for 10 min at 5000× *g*. The supernatant was discarded and the bacterial pellet was resuspended in 4.5 mL LB medium. The cell solutions (0.5 mL) were mixed with 4.5 mL sterile seaweed alginate solution (60% guluronic acid content) for production of beads or mix 0.5 mL 0.9% NaCl with 4.5 mL sterile seaweed alginate solution as control. The mixture was forced once with air through a cannula into a solution (0.1 M Tris-HCl containing 0.1 M CaCl_2_, pH 7.0). The suspension containing 6 × 10^9^ CFU/mL was confirmed by colony counts.

Before challenge with bacteria, all rats were anaesthetized with pentobarbital (10 mL/kg body weight). With a bead tipped needle, intratracheal challenge with 50 µL of *P. aeruginosa* (6 × l0^9^ CFU/mL) embedded in alginate beads or 0.9% NaCl embedded in alginate beads as negative control was performed. The inoculum was installed in the left lung 11 mm from the tracheal penetration site and the incision was sutured with silk. After the infection, the mice were monitored during their recovery, and the wounds healed without any complications. From the second day after challenge, rats in each group received drugs or placebo once a day. 

The day after infections, rats challenged with *P. aeruginosa* were randomized into three groups. (i) Baicalin-treated group: Baicalin solution (5 mg/mL) was injected intraperitoneally using a dosage of 5 mg/kg of body weight once a day. The dosage was decided on the basis of the reported baicalin dosage pilot study. (ii) Cefepime-treated group: Cefepime Hydrochloride for Injection (Shandong qilu, Zibo, China), a commercial product which is commonly used as an anti-infection agent, was used as a control drug in the study. Cefepime was administered intraperitoneally at a dosage of 40 mg/kg of body weight once a day. The dosage was determined on the basis of the dosage used in the treatment of infection. (iii) Model group: Rats were received an injection of same amount of sterile PBS intraperitoneally once a day. (iv) Control group: Rats were received an injection of same amount of sterile saline intraperitoneally once a day. Four days after challenge, blood was drawn from abdominal aorta. All rats were sacrificed by using 20% pentobarbital using a dosage of 5 mg/kg of body weight. 

### 4.10. Macroscopic Description of the Lungs

According to the severity of the infection, all lungs were assigned to one of four levels as described by Johansen et al. [59]. The scores are recorded as follow: with 1, being normal; 2, swollen lungs, hyperemia, small atelectasis (1 by 1 mm); 3, adherences, small hemorrhages, small abscesses (to 1 by 2 mm), atelectasis (2 by 3 mm); 4, adherences, hemorrhages, abscesses (>1 by 2mm), and atelectasis (>3 mm) [59]. The scoring was performed in a blinded fashion to avoid bias.

### 4.11. Histopathological Studies and Quantitative Bacteriology

The lower section of the left lung lobe was fixed in formalin for at least 1 week, and then paraffin embedded, cut into 5 to 10-jim-thick sections, and stained with hematoxylin and eosin (HE).

The lungs of the remaining half of the animals in each group were prepared for the quantitative bacteriological examination. The lungs were removed aseptically from mice and homogenized in 5 mL physiological saline, followed by centrifuging at 1500× *g* for 15 min. The supernatants were subjected to tenfold serial dilutions. Subsequently, the proper dilutions were plated in triplicate in PIA plates and the CFU per lung was calculated. 

### 4.12. ELISA

All animals were anaesthetized. Blood was drawn from abdominal aorta. Concentrations of TNF-α were measured by enzyme-linked immunosorbent assay (ELISA) using kit available from Beijing Keyingmei. 

### 4.13. Statistical Analysis

All data were analyzed by Student’s *t* test except those from the animal studies. The Mann-Whitney U test was used for analysis of nonparametric data to determine the significance of the differences in the numbers of CFU between two groups of rats and categorical data were compared by the chi-square test. SPSS software (version 16.0, Chicago, IL, USA) was used for statistical analysis. A *p* value of <0.05 was considered statistically significant. 

## 5. Patents 

A patent entitled “Use of baicalin for treating and preventing bacterial infections” has been granted by State Intellectual Property Office of PR China (ZL 2010 1 0202023.2), which is associated with the results reported in this paper.

## Figures and Tables

**Figure 1 molecules-26-01497-f001:**
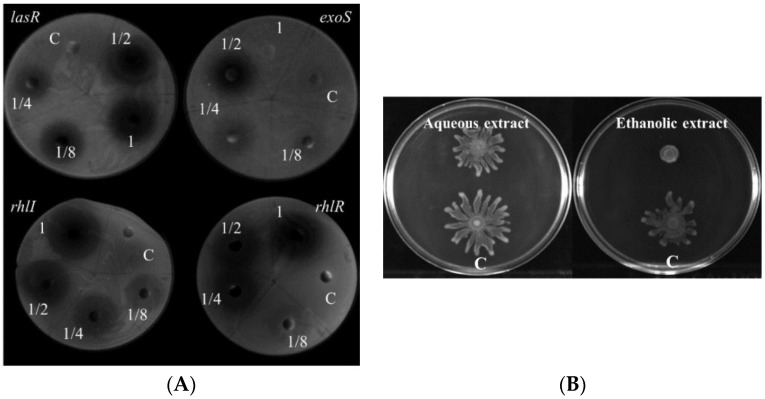
Inhibition of virulence genes expression and swarming motility by the extract of *Scutellariae radix*. (**A**) Inhibition on the expression of *lasR*, *exoS*, *rhlI* and *rhlR* by the ethanolic extract of *Scutellariae radix*. 1, 1/2, 1/4 and 1/8 represent the concentrations of exacts at 172.5 mg/mL, 86.25 mg/mL, 43.13 mg/mL, 21.56 mg/mL respectively; C, solvent as control. (**B**) Inhibition of swarming motility in *P. aeruginosa* PAO1 by the aqueous and ethanolic extracts. The concentration of extracts was 43.13 mg/mL.

**Figure 2 molecules-26-01497-f002:**
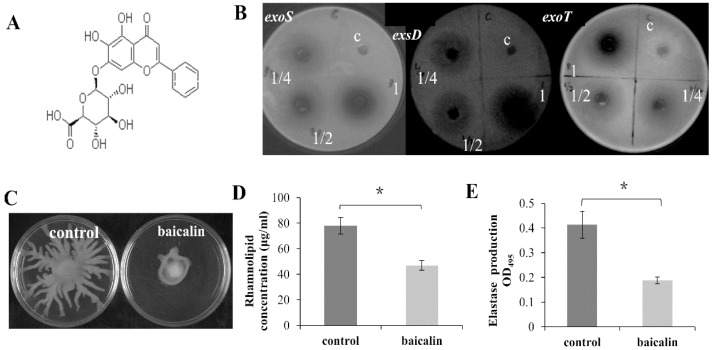
Repression on the virulence genes expression and phenotypes in *P. aeruginosa* by baicalin (**A**) The chemical structure of baicalin. (**B**) Inhibition on the expression of T3SS genes *exoS*, *exsD*, and *exoT* by baicalin. Five microliters of baicalin solution was dropped on the plates. 1, 1/2 and 1/4 represent that the concentrations of baicalin are 25, 12.5 and 6.25 mg/mL respectively; C, solvent as control. (**C**) Repression of swarming motility bybaicalin. Baicalin was added at 250 µg/mL. Inhibition of rhamnolipid production (**D**) and elastase production (**E**) by baicalin. Baicalin was added at 250 µg/mL. Averages of triplicate experiments ± standard errors of the means are shown. The results shown are representative of three independent experiments with similar trends. * significant difference (*p* < 0.05).

**Figure 3 molecules-26-01497-f003:**
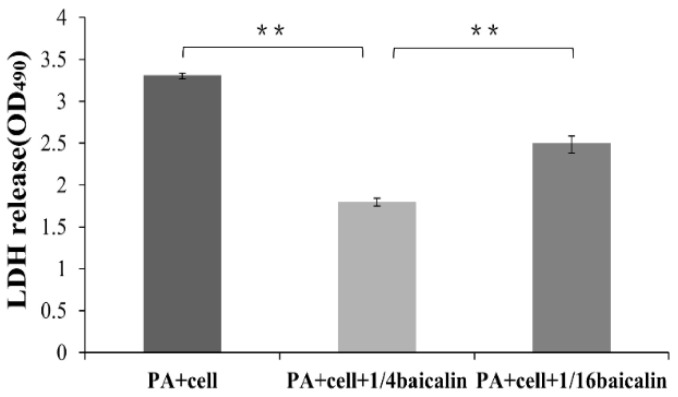
Inhibition of cytotoxicity of *P. aeruginosa* PAO1 to eukaryotic cells by baicalin. 1/4 and 1/16 represent 250 μg/mL and 62.5 μg/mL of baicalin respectively; Averages of triplicate experiments ± standard errors of the means are shown. The results shown are representative of three independent experiments with similar trends. ** very significant difference (*p* < 0.01).

**Figure 4 molecules-26-01497-f004:**
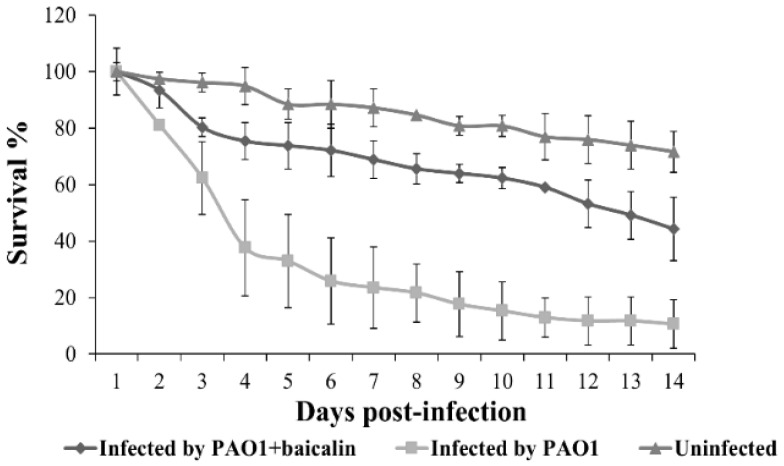
Inhibition of pathogenicity of PAO1 by baicalin in fly infection model. Baicalin was added at 250 µg/mL. Averages of three independent experiments ± standard errors of the means are shown.

**Figure 5 molecules-26-01497-f005:**
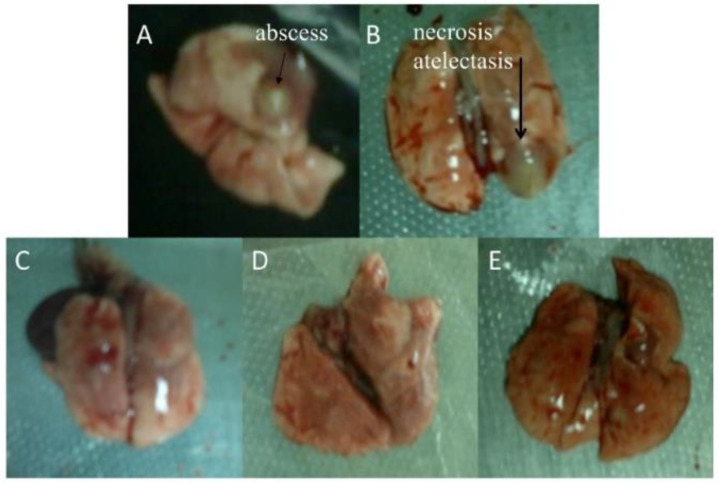
Morphological pathology of rat lungs in different experimental groups. (**A**,**B**) Panels represent the lungs of rats in Model group. (**C**–**E**) represent the blank Control group, Cefepime-treated group and Baicalin-treated group.

**Figure 6 molecules-26-01497-f006:**
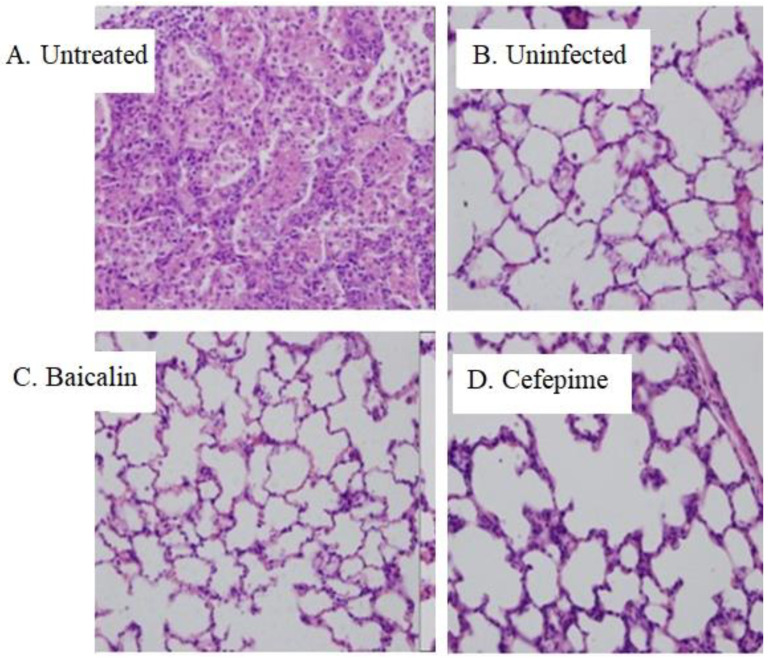
Microscopic changes of the lung tissues of the different animal groups (H&E staining, magnification: ×200). Panels (**A**–**D**) show pathology of Model group, Blank uninfected control group, Baicalin-treated group and Cefepime-treated group respectively. (**A**) lung hyperemia, large amount of inflammatory cell infiltration, vascular and alveolar walls dilatation; (**B**) light lung hyperemia; (**C**) milder lung hyperemia and few inflammatory cell infiltrations; (**D**) alveolar walls widened, few inflammatory cell infiltration and milder lung hyperemia.

**Figure 7 molecules-26-01497-f007:**
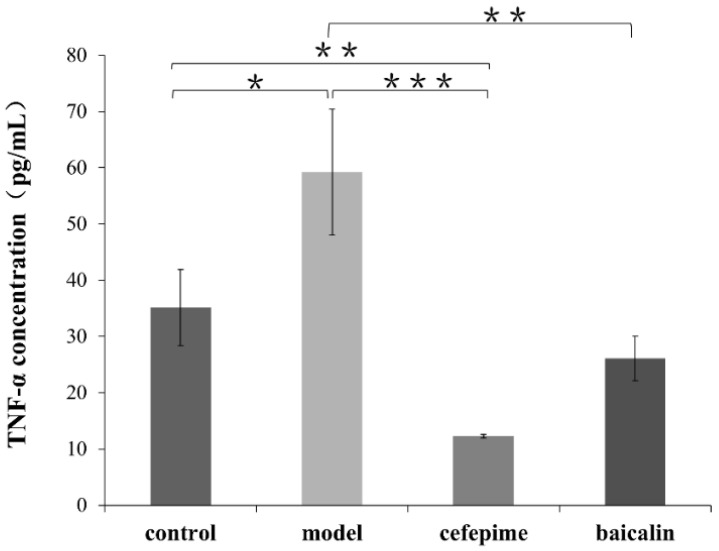
Levels of TNF-α in serum of rats in different groups. Averages of triplicate experiments ± standard errors of the means are shown. The results shown are representative of three independent experiments with similar trends. * significant difference (*p* < 0.05); ** very significant differences (*p* < 0.01); *** highly significant differences (*p* < 0.001).

**Figure 8 molecules-26-01497-f008:**
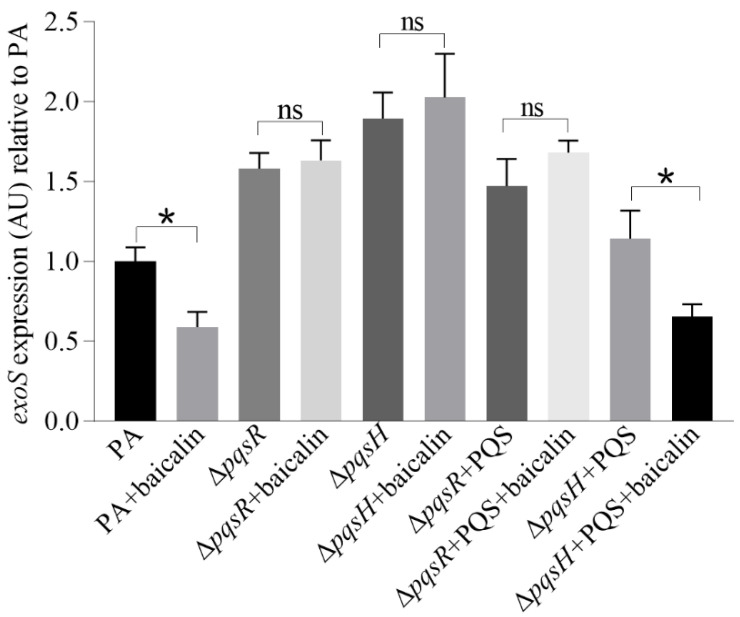
Relative expression levels of *exoS* in PAO1, Δ*pqsR* and Δ*pqsH* under different conditions. The activity of *exoS* promoter in soft agar medium was measured as described in Materials and Methods. Baicalin was added at 250 µg/mL. PQS was added at 1 µg/mL. After overnight incubation at 37 °C, light production of the reporter strain was recorded by a LAS-3000 imaging system (Fuji Corp, Tokoyo, Japan). The relative promoter activity was evaluated by the software multi gauge version 3.0. AU strands for Arbitrary Unit, a unit to measure the amount of chemiluminescence recorded by LAS-3000. It signifies the relative density value accumulated as linear data by a CCD camera in the image surface. Averages of triplicate experiments ± standard errors of the means are shown. The results shown are representative of three independent experiments with similar trends. * significant difference (*p* < 0.05) ns, no significant difference.

**Table 1 molecules-26-01497-t001:** Microscopic scoring of the lung pathology in different groups.

Treatment Group	Scoring of Lungs According to Severity of Inflammation Maroscopic Mean
NaCl	1.1 *** (*p* = 0.0002)
Cefepime	1.3 * (*p* = 0.01)
Model	1.9
Baicalin	1.4 * (*p* = 0.02)

* significant difference (*p* < 0.05) compared to the Model group; *** highly significant differences (*p* < 0.001) compared to the Model group (Chi-square test, χ^2^).

**Table 2 molecules-26-01497-t002:** The numbers of CFU of *P. aeruginosa* in rat lungs.

Treatment Group	Range of log.CFU. per Organ (CFU) (Median)
Cefepime	(0–1.1) 0 ** (*p* = 0.001)
Untreated model	(0–5.5) 3.1
Baicalin	(0–3.3) 0 * (*p* = 0.04)

* significant difference (*p* < 0.05) compared to the Model group; ** very significant differences (*p* < 0.01) compared to the Model group (Mann–Whitney U test).

**Table 3 molecules-26-01497-t003:** Bacterial strains and plasmids used in this study.

Strains or Plasmids	Relevant Characteristics	Source or Reference
Strains		
PAO1	Wild type	
PAO1(Δ*pqsR*)	*pqsR* mutation of PAO1	[28]
PAO1(Δ*pqsH*)	*pqsH* mutation of PAO1	This study
PAO1(Δ*retS*)	*retS* mutation of PAO1	[44]
PAO1(Δ*gacA*)	*gacA* mutation of PAO1	[44]
PAO214	*lasI* mutation of PAO1	[45]
PDO100	*rhlI* mutation of PAO1	[46]
Plasmids		
pMS402	Expression reporter plasmid with the promoterless *luxCDABE* gene; Kn^r^, Tmp^r^	[47]
CTX6.1	Integration plasmid with the mini-CTX-*lux* backbone; Tc^r^	This lab
pRK2013	Helper vector in triparental conjugation; Tra^+^, Kn^r^	[48]
pkD-*pilG*	*pilG* promoter was inserted into pMS402; Kn^r^, Tmp^r^	[47]
pkD-*fliC*	*fliC* promoter was inserted into pMS402; Kn^r^, Tmp^r^	[47]
pkD-*rhlA*	*rhlA* promoter was inserted into pMS402; Kn^r^, Tmp^r^	[47]
pkD-*phzA1*	*phzA1*promoter was inserted into pMS402; Kn^r^, Tmp^r^	[47]
pkD-*phzA2*	*phzA2* promoter was inserted into pMS402; Kn^r^, Tmp^r^	[47]
pkD-*lasI*	*lasI* promoter was inserted into pMS402; Kn^r^, Tmp^r^	[49]
pkD-*rhlI*	*rhlI* promoter was inserted into pMS402; Kn^r^, Tmp^r^	[49]
pkD-*lasR*	*lasR* promoter was inserted into pMS402; Kn^r^, Tmp^r^	[49]
pkD-*rhlR*	*rhlR* promoter was inserted into pMS402; Kn^r^, Tmp^r^	[49]
pkD-*rhlA*	*rhlA* promoter was inserted into pMS402; Kn^r^, Tmp^r^	[47]
pKD-*lasA*	*lasA* promoter was inserted into pMS402; Kn^r^, Tmp^r^	[47]
pkD-*lasB*	*lasB* promoter was inserted into pMS402; Kn^r^, Tmp^r^	[47]
pkD-*rsmA*	*rsmA* promoter was inserted into pMS402; Kn^r^, Tmp^r^	This lab
pkD-*rsmY*	*rsmY* promoter was inserted into pMS402; Kn^r^, Tmp^r^	This lab
pkD-*rsmZ*	*rsmZ* promoter was inserted into pMS402; Kn^r^, Tmp^r^	This lab
pKD-*gacA*	*gacA* promoter was inserted into pMS402; Kn^r^, Tmp^r^	[44]
pKD-*rpoS*	*rpoS* promoter was inserted into pMS402; Kn^r^, Tmp^r^	[47]
pkD-*pilG*	*pilG* promoter was inserted into pMS402; Kn^r^, Tmp^r^	[47]
pkD-*fliC*	*fliC* promoter was inserted into pMS402; Kn^r^, Tmp^r^	[47]
pKD-*pqsA*	*pqsA* promoter was inserted into pMS402; Kn^r^, Tmp^r^	[50]
pkD-*pqsR*	*pqsR* promoter was inserted into pMS402; Kn^r^, Tmp^r^	[50]
pkD-*pqsH*	*pqsH* promoter was inserted into pMS402; Kn^r^, Tmp^r^	[28]
pkD-*oprH*	*oprH* promoter was inserted into pMS402; Kn^r^, Tmp^r^	[47]
pkD-*exoS*	*exoS* promoter was inserted into pMS402; Kn^r^, Tmp^r^	[47]
pkD-*exoY*	*exoY* promoter was inserted into pMS402; Kn^r^, Tmp^r^	[47]
pkD-*exoT*	*exoT* promoter was inserted into pMS402; Kn^r^, Tmp^r^	[47]
pkD-*exsD*	*exsD* promoter was inserted into pMS402; Kn^r^, Tmp^r^	[44]
pkD-*exsC*	*exsC* promoter was inserted into pMS402; Kn^r^, Tmp^r^	[44]
pkD-*aprA*	*aprA* promoter was inserted into pMS402; Kn^r^, Tmp^r^	[47]
pkD-*migA*	*migA* promoter was inserted into pMS402; Kn^r^, Tmp^r^	[47]
pkD-*oprH*	*oprH* promoter was inserted into pMS402; Kn^r^, Tmp^r^	[47]
pkD-*rnr*	*rnr* promoter was inserted into pMS402; Kn^r^, Tmp^r^	[47]
pkD-*xcpR*	*xcpR* promoter was inserted into pMS402; Kn^r^, Tmp^r^	[47]
*ctx-exoS*	Integration plasmid, *exoS* promoter with promoterless *luxCDABE* gene was inserted into CTX6.1; Kn^r^, Tmp^r^, Tc^r^	This lab
ctx-*phzA1*	Integration plasmid, *phzA1* promoter with promoterless *luxCDABE* gene was inserted into CTX6.1; Kn^r^, Tmp^r^, Tc^r^	This lab
ctx-*phzA2*	Integration plasmid, *phzA2* promoter with promoterless *luxCDABE* gene was inserted into CTX6.1; Kn^r^, Tmp^r^, Tc^r^	This lab
ctx-*vfr*	Integration plasmid, *vfr* promoter with promoterless *luxCDABE* gene was inserted into CTX6.1; Kn^r^, Tmp^r^, Tc^r^	This lab

## Data Availability

The data presented in this study is available within the article.

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
