# Peer review of "Baicalin Represses Type Three Secretion System of Pseudomonas aeruginosa through PQS System"

_molecules, 2021, doi:10.3390/molecules26061497_

Round 1

Reviewer 1 Report

In manuscript molecules-1129755, the authors have screened Chinese traditional medicines for anti-virulence activity against Pseudomonas aeruginosa, rather than bacterial viability screening.  This led to investigating Scutellariae radix and its major component baicalin.  These workers demonstrated that baicalin diminished several virulence factors in P. aeruginosa.  In addition, baicalin reduced in-vitro cytotoxicity of P. aeruginosa on mammalian cells, and reduced in-vivo pathogenicity in Drosophila melanogaster and Sprague-Dawley rat lungs.

This work has demonstrated that baicalin should be pursued as a complementary treatment option for Pseudomonas infections.  Publication is warranted and recommended.

Section 4.2:  What was the source of Scutellariae radix?  Is it a commercial source or was it harvested?  What species of Scutellaria is used for Scutellariae radix?

There are minor English corrections needed throughout the manuscript.  For the most part, I had no problem understanding the text, but the manuscript would benefit by proof-reading by a native speaker of English.

Reviewer 2 Report

The manuscript describes using Baicalin extracted from Scutellariae radix as such an active anti-virulence compound against Pseudomonas aeruginosa. Baicalin is shown to reduce severity of infection as well as decrease virulence factor production in PAO1.  Overall, the study is well written but there are some issues with grammar that can easily be fixed with editing. Only minor edits are requested by this reviewer. Please see below:

 Concerns:

The middle image in Figure 2B is very poor quality and can barely be seen. This needs to be redone and replaced with a higher quality image.

Figure 2 D&E, Figure 3, Figure 4, Figure 7 and Figure 8: specify what the errors bars represent and how many replicates were used to generate the data

Figure 6 is missing magnification

Specific comments:

Line 23: both in vitro and in vivo should be in italics

Line 26: in vivo should be in italics

Line 28: Pseudomonas should be in italics

Line 41: fund should be found

Line 46: use "successful" instead of triumphant 

Line 56: indicate should be indicates

Line 72-73: replace "They hence" with "Hence they"

Line 95: replace "on a plenty of" with "on numerous"

Line 330: Define CFU at first use
